# Translation, Cross-Cultural Adaptation and Psychometric Validation of the Arabic Version of the Cardiac Rehabilitation Barriers Scale (CRBS-A) with Strategies to Mitigate Barriers

**DOI:** 10.3390/healthcare11081196

**Published:** 2023-04-21

**Authors:** Raghdah Aljehani, Sherry L. Grace, Aseel Aburub, Karam Turk-Adawi, Gabriela Lima de Melo Ghisi

**Affiliations:** 1Rehabilitation Department, King Abdullah Medical City, Makkah 24246, Saudi Arabia; 2Faculty of Health, York University, Toronto, ON M3J 1P3, Canada; 3KITE, Toronto Rehabilitation Institute, University Health Network, Toronto, ON M4G 2V6, Canada; 4Peter Munk Cardiac Centre, University Health Network, University of Toronto, Toronto, ON M5G 2C4, Canada; 5Department of Physiotherapy, Applied Science Private University, Amman 11931, Jordan; 6College of Health Sciences, QU Health, Qatar University, Doha 2713, Qatar

**Keywords:** cardiac rehabilitation, questionnaires and surveys, psychometrics, validity (epidemiology), access to health care, referral

## Abstract

Cardiac rehabilitation (CR) utilization is low, particularly in Arabic-speaking countries. This study aimed to translate and psychometrically validate the CR Barriers Scale in Arabic (CRBS-A), as well as strategies to mitigate them. The CRBS was translated by two bilingual health professionals independently, followed by back-translation. Next, 19 healthcare providers, followed by 19 patients rated the face and content validity (CV) of the pre-final versions, providing input to improve cross-cultural applicability. Then, 207 patients from Saudi Arabia and Jordan completed the CRBS-A, and factor structure, internal consistency, construct, and criterion validity were assessed. Helpfulness of mitigation strategies was also assessed. For experts, item and scale CV indices were 0.8–1.0 and 0.9, respectively. For patients, item clarity and mitigation helpfulness scores were 4.5 ± 0.1 and 4.3 ± 0.1/5, respectively. Minor edits were made. For the test of structural validity, four factors were extracted: time conflicts/lack of perceived need and excuses; preference to self-manage; logistical problems; and health system issues and comorbidities. Total CRBS-A α was 0.90. Construct validity was supported by a trend for an association of total CRBS with financial insecurity regarding healthcare. Total CRBS-A scores were significantly lower in patients who were referred to CR (2.8 ± 0.6) vs. those who were not (3.6 ± 0.8), confirming criterion validity (*p* = 0.04). Mitigation strategies were considered very helpful (mean = 4.2 ± 0.8/5). The CRBS-A is reliable and valid. It can support identification of top barriers to CR participation at multiple levels, and then strategies for mitigating them can be implemented.

## 1. Introduction

Cardiovascular diseases (CVDs) remain a leading contributor to morbidity and mortality worldwide [1]. The burden of CVD is particularly high in countries where Arabic is the official language (i.e., Arabic countries) [2]; globally, this comprises 22 countries. For instance, data from the Eastern Mediterranean (EMR)/the Middle East and North African (MENA) regions—where the majority of countries are comprised of people whose first language is Arabic—show high rates of disability related to CVD, and its incidence is forecasted to grow fastest in these two regions [3,4]. Thus, there is a great need for secondary and tertiary prevention to mitigate the CVD burden in the region.

Cardiac rehabilitation (CR) is such a comprehensive outpatient chronic disease management model with well-established benefits, including reduced hospital admissions and CVD mortality rates [5]. There is evidence demonstrating that CR is effective in Arabic countries as well [6]. Despite these benefits, CR utilization is low around the world [7], including in these regions [8,9].

Barriers are known to exist at the health system, referring provider, program, and patient levels [10]. Grace et al. have developed the Cardiac Rehab Barriers Scale (CRBS) assessing patients’ perceptions of these multi-level barriers to CR enrollment and participation [11,12]. The CRBS has been translated to 17 languages, including 4/5 of the most spoken languages in the world by number of native speakers (Chinese 1.3 billion, Spanish 475 million, English 373 million, and Hindi 344 million) [13]. Arabic is the fourth, with 362 million native speakers around the world [13]. Given the socio-cultural context in Arabic countries, there may be some unique barriers. The scale is key to identifying barriers so they can be mitigated [14]. Indeed, strategies to mitigate top identified barriers could be provided to patients and have recently been expounded.

Accordingly, the aims of this study were to: (a) rigorously translate and cross-culturally adapt the CRBS to Arabic by using the best practices, and then (b) psychometrically validate the translation. This included factor structure, internal reliability, as well as the following forms of validity: face, content, cross-cultural, construct, and criterion. The secondary objectives were to (a) translate and cross-culturally adapt potential strategies to mitigate the barriers, and (b) solicit patient input on their usefulness.

## 2. Materials and Methods

### 2.1. Design

This was a multi-method study comprising (1) translation and cultural adaptation of the CRBS to Arabic using best practices [15], followed by (2) a cross-sectional survey of patients for psychometric validation. Figure 1 displays the multi-step translation, cross-cultural adaptation, and psychometric validation processes (dates provided). The study was approved by the King Abdullah Medical City (22-988; Saudi Arabia) and York University (e2021-013; Canada) research ethics boards.

### 2.2. Materials: Cardiac Rehabilitation Barriers Scale and Barrier Mitigation Strategies

The CRBS is a patient-report survey developed by Grace et al. to assess their perceptions of patient, referring provider, CR program and health system-level barriers to phase II CR enrolment and adherence [11], so reasons for under-use can be understood and addressed [16]. It was developed following an extensive review of the literature, with feedback from cardiologists and CR staff, followed by validation in patient samples; validity and reliability have been established [12]. 

The original English version of the scale is comprised of 21 items, each scored on a 5-point Likert scale (1 = strongly disagree to 5 = strongly agree), as well as an open-ended item for additional barriers. Higher scores indicate greater barriers to CR. The original scale and most translations consist of four subscales: perceived need/healthcare factors; logistical factors; work/time conflicts; and comorbidities/functional status [12]. 

The revised version of the CRBS (i.e., CRBS-R) was used in this study for the first time (https://sgrace.info.yorku.ca/cr-barriers-scale/crbs-instructions-and-languages-translations/, accessed on 8 March 2023), relevant to supervised and unsupervised CR [12]. It was developed following a review of the CRBS literature (45 theses, abstracts, and papers from 25 countries), including psychometric properties, greatest and lowest barriers, as well as additional barriers identified in the studies [12]. Clarifying edits were made to the instructions, and a ‘not applicable’ response option added for each of the items. Item changes primarily involved explication of examples for some of the barriers based on the additional barriers identified in the literature, and re-ordering of items to group related barriers together.

Mitigation responses for each barrier were created by co-author SG, based on a review of quality improvement strategies in the CR field [17] (see Appendix A). Where patients rate a barrier as 4 or 5 out of 5, it is recommended that the mitigation strategy is proffered [12]. These were pilot-tested in English, Simplified Chinese, and Portuguese prior to this study (https://globalcardiacrehab.com/Patients-CRBS, accessed on 8 March 2023). 

### 2.3. Phase 1: Multi-Step Process for Arabic Translation and Cultural Adaptation of CRBS and Associated Barrier Mitigation Strategies

The initial translation of the scale and mitigation strategies from English to the target language (Arabic) was performed by two co-authors (RA and AA) independently; both are fluent in English, and their mother tongue is Arabic. After the two translations had been performed, three co-authors (RA, AA, and KTA) combined and considered the wording. This first version of the instrument was then back-translated into English by co-author KTA, which was reconciled with the merged forward translations by the three Arabic-speaking, bilingual authors, resulting in the second version.

Next, in October 2022, a review committee was formed, comprising a convenience sample of Arabic-speaking healthcare providers who were experts in the field of CVD (including physiotherapists, cardiologists, physiatrists, and family physicians). Contacts from across the EMR were invited to participate via email by the Arabic-speaking co-authors and from the International Council of Cardiovascular Prevention and Rehabilitation community [18]. Using Qualtrics (https://www.qualtrics.com/, accessed on 8 March 2023), they were asked to rate the content validity of items (to enable computation of the content validity index (CVI) for the items (I-CVI) and scale (S-CVI) [19] and usefulness of mitigating strategies. Specifically, for each barrier, respondents were asked ‘yes’ or ‘no’ whether any changes to the item were needed to ensure cross-cultural relevance to Arabic patients, or to improve semantic clarity of the items. If yes, an open-ended description was solicited. They were also asked whether any other barriers should be added. Ratings for mitigation responses ranged from 1 = very unhelpful to 5 = very helpful, with the option to enter open-ended input for each. Edits were made accordingly. 

Next, CR-eligible in or out-patients [20] with a diagnosis of heart disease who were or were not enrolled in CR were purposively-sampled, including to ensure representation of diverse ages, sexes and socioeconomic backgrounds, as well as in clinical presentation. They were asked to review the revised version of the scale and corresponding mitigation strategies in November 2022. Recruitment sources are outlined below. Exclusion criteria were: younger than 18 years old, inability to understand Arabic, and having any significant visual or cognitive condition, or serious mental illness which would preclude their ability to answer the questionnaire. The target sample size was based on the International Society for Pharmacoeconomics and Outcomes Research (ISPOR) guidelines [15], which suggest a minimum of 5–8 respondents.

For this phase, again using Qualtrics, participants were asked to rate clarity of each item from 1 = very unclear to 5 = very clear, or to denote that the item is ‘not applicable’ to them. They were also asked to report any additional barriers. For each corresponding mitigation strategy, participants were asked to rate how useful the response would be if the barrier were applicable to them, on a scale from 1 = not at all useful to 5 = extremely useful. There was also an open-ended option for participants to offer suggestions on how to improve each response. Edits were considered based on responses.

Finally, the back-translated and input-revised version of the Arabic scale was then compared with the original version to consider conceptual discrepancies, with revisions again considered. Overall, this step assessed face, content, and cross-cultural validity of the CRBS-A by multiple stakeholders, and the drafted CRBS-A was ready for psychometric validation.

### 2.4. Phase 2: Cross-Sectional Survey of Patients for Psychometric Validation

#### 2.4.1. Procedure

Participants were recruited in December 2022 and January 2023. Participants gave online consent and completed the survey in Qualtrics. Psychometric properties assessed were factor structure and internal consistency, as well as construct and criterion validity. It was hypothesized that referred patients would have fewer barriers.

#### 2.4.2. Setting and Participants

Inclusion and exclusion criteria for patients are outlined above. Participants recruited for the initial scale review and ultimate psychometric validation were recruited at multiple healthcare centers in Saudi Arabia (where they have CR programs) and in Jordan (where there is no CR, although patients have access to physical therapy), given CR is not pervasive in Arabic-speaking countries [8]. Centers in Saudi Arabia were King Abdullah Medical City (Makkah), Al Noor Hospital (Makkah), and Prince Sultan Cardiac Centre (Riyadh). Centers in Jordan were the King Abdullah University Hospital (Irbid) and Albasheer Hospital (Amman). The patient inclusion and exclusion criteria from step 1 applied. For the psychometric validation phase, to ensure adequate sample size for the factor analysis, 10 patients were sought per item [21], or in this case, 210 participants.

#### 2.4.3. Measures

Sociodemographic characteristics (e.g., age, sex, socioeconomic status, social support) were self-reported using investigator-generated items with forced-choice response options. To assess criterion validity, if they were referred to CR (yes/no) was collected via self-report as well.

Upon completing the survey, corresponding mitigating strategies were provided to patients for any barrier scored ≥ 4/5. After reviewing them, participants were asked how helpful these were on a Likert-type scale ranging from 1 = very unhelpful to 5 = very helpful.

### 2.5. Data Analysis

The Statistical Package for Social Sciences v.28 (SPSS Inc., Chicago, IL, USA) was used for all data analysis. Descriptives of responses from the expert and patient review were examined by the co-authors, with edits made to items as applicable as outlined above. 

For the patient validation survey, first, a descriptive analysis of participant characteristics was performed. Next, exploratory factor analysis (EFA) was undertaken. Factor extraction was conducted using the principal component method, with varimax rotation. The number of factors extracted was determined by considering those with eigenvalues ≥ 1.0, percentage of variance accounted for, and examination of the Scree plot. Item factor loadings ≥ 0.3 were considered in finalizing the items for each factor and interpreting them.

A descriptive examination was performed of each CRBS item. Where participants completed more than 80% of the items, a mean total CRBS score was computed. Subscale scores were also computed based on the results of the factor analysis. 

Next, internal consistency was determined by calculating Cronbach’s *alpha* values of the scale and subscales. *Alpha* ≥ 0.70 was considered acceptable, reflecting the correlation of the items among themselves and with the total score [21]. The scale’s Cronbach’s alpha reliability coefficient for internal consistency if the individual item is removed from the scale was also checked.

The CRBS was not normally distributed. Therefore, to assess construct validity, Spearman’s correlation, Wilcoxon tests, and Kruskal–Wallis tests were applied to explore associations between sociodemographic characteristics of study participants and their CRBS scores, given associations between CR use and social determinants of health [22]. Finally, to consider criterion validity, differences in total CRBS scores by CR referral were tested with Wilcoxon tests. Note that given there was no available CR in Jordan and the number of tests which can inflate error, construct validity assessments and of the association between the CRBS items and CR enrolment were undertaken on an exploratory basis only. The level of significance for all tests was set at 0.05. 

## 3. Results

The translation and cultural adaptation process as well as psychometric validation proceeded as planned. This is outlined below. 

### 3.1. Translation and Cultural Adaptation

Following translations and harmonization of the CRBS to Arabic, the 19 expert health professionals deemed all 21 items in the original CRBS version applicable to the Arabic context. For items 1 and 3, two respondents suggested changes; for item 2, three suggested changes, and for items 6, 7, and 20, one respondent each suggested change. Minor changes to the instructions and in items 1, 3, 5, and 6 were made based on these comments from experts (Appendix B). The I-CVIs ranged from 0.8 to 1.0, and the S-CVI was 0.9, which establishes that the Arabic version of CRBS has acceptable content validity.

Helpfulness ratings of barrier mitigation responses ranged from 4.1 to 4.4/5 (mean 4.3 ± 0.1). Given this and in reviewing the open-ended input, only minor edits were made to mitigation strategy responses. 

For the patients’ input, 19 reviewed the CRBS-A items and responses. Clarity of items ranged from 4.3 to 4.7/5 (mean 4.5 ± 0.1). The number of respondents selecting “not applicable” for each item ranged from 1 to 3 (i.e., maximum 16% of respondents). In addition, considering open-ended input, no further changes were made to the CRBS-A at this stage. The final CRBS-A is shown in Appendix B.

Patient helpfulness of response ratings ranged from 4.3 to 4.7/5 (mean 4.5 ± 0.1). Given the input (which included comments such as “very clear”), no edits were made to mitigation strategy responses (Appendix A).

### 3.2. Psychometric Validation

The sample was comprised of 207 participants, of which 49 (23.7%) participated in CR. Most patients were male, retired, with a mean age of 58, and with 13 years of formal education (Table 1).

The structure of the scale was first assessed using EFA. The Kaiser–Meyer–Olkin value was 0.875, and Bartlett’s test was significant (chi-square 2270.417; *p* < 0.001), which indicated data suitability for factor analysis. Four components with eigenvalues ≥ 1.0 were obtained. These factors, considered together, accounted for 62.7% of the total variance. Table 2 displays the factor structure of the CRBS-A, including item loadings.

The first factor had 8 items and reflects time conflicts and a lack of perceived need/excuses. The second factor (5 items) reflects preference to self-manage. The third factor (3 items) reflects logistical barriers. The fourth factor (5 items) reflects health system issues and comorbidities. The first three factors of the Arabic version of the CRBS showed good internal consistency (Cronbach′s *alpha* ≥ 0.7; Table 2). However, the alpha for factor 4 fell short of the 0.7 threshold. Overall, however, as the α for the total CRBS was 0.90, the internal consistency of the scale is supported.

As shown in Table 1, with regard to construct validity, there was only a trend for an association between total barriers and greater worry related to healthcare costs. With regard to criterion validity, significant associations were observed between total scores and being referred to CR.

Finally, top barriers for those that enrolled in CR were “distance”, “difficulties in accessing sessions that require attendance in person”, and “it took too long to get referred and into the program”. Top barriers for those that did not were “I did not know about cardiac rehab”, “my doctor did not feel it was necessary”, and “difficulties in accessing sessions that require attendance in person”. Exploratory analyses were performed to examine CRBS-A item, subscale, overall scores by CR enrolment. Respondents who enrolled in CR had significantly lower scores on 4/21 items (cost, not knowing about CR, doctor not feeling it was necessary, many people with heart problems do not go and they are fine) and higher on 6/21 items (already exercising at home, severe weather, travel, time constraints, work responsibilities, too long to get referred) than those who did not. While there was no difference in total scores by enrolment status, factor 4 subscale scores (i.e., health system issues, comorbidities) were significantly greater in those who did not enroll (*p* = 0.04), with a trend for factor 3 (logistical barriers; *p* = 0.08).

### 3.3. Main Barriers and Usefulness of Strategies to Mitigate These Barriers

CRBS item scores ranged from 2.2 to 4.1/5 (Table 3). The top CR barriers were “I did not know about cardiac rehab”, “my doctor did not feel it was necessary”, “distance”, and “difficulties in accessing sessions that require attendance in person”. Items with the highest number of participants indicating that the barrier was not applicable were: “other health problems prevent me from going” (16% of respondents), “work responsibilities” (15%), and “I prefer to take care of my health alone, not in a group” (14%). Other barriers cited by participants included no knowledge of what CR is and not having CR centers close to home, which are already included items (items 5 and 1, respectively).

Overall, the mean usefulness of the information provided to mitigate the barriers was rated 4.2 ± 0.8/5; 85% of respondents found the information to be useful or very useful.

## 4. Discussion

Following best practices, this study has rigorously translated, cross-culturally adapted, and psychometrically validated the Arabic version of the CRBS scale. Through this process, all 21 items of the scale were retained, with revisions made to 4 items to improve clarity based on expert and patient reviews. Face, content, and cross-cultural validity were supported. Subsequent exploratory factor analysis revealed four factors: time conflicts, lack of perceived need and excuses; preference to self-manage; logistical problems; as well as health system issues and comorbidities. Even in the EMR context of low CR availability, criterion validity was confirmed by the significantly lower mean CRBS scores in patients who were referred to CR than those who were not. Overall, these results confirm the validity and reliability of the CRBS-A in EMR settings with and without accessible CR. This study has also established for the first time mitigation strategies for each barrier (Appendix A), which patients rated as highly useful. 

It is important to compare and contrast this CRBS translation with others. In fact, this is the 18th translation of the CRBS; among all versions, there are seven that also comprise four factors. However, the items that loaded on each factor were somewhat different, potentially due to the fact that the translation is based on the CRBS-R. The internal consistency of the Arabic version was 0.90, which is one of the highest when compared to all other CRBS translations, again potentially due to the fact that it is based on the CRBS-R. The reliability of some subscales in seven of the 18 CRBS translations was below 0.70 [21], similar to factor 4 in this Arabic translation; this is likely due to the fact that some barriers are quite unique. The reader is referred to the review of all CRBS translations for more information [12]. 

Mean scores were comparable to other samples which consist of many non-enrollees [12], as well as with other samples from the EMR [9,23,24]. Indeed, the CRBS has now been administered in four EMR countries (i.e., Qatar [9], Iran [23,24], Saudi Arabia, Jordan), establishing that it is valid in the socio-religious context as well. The CRBS has been administered in other countries with very low CR availability [25] such as Brazil [26,27,28,29,30,31], Colombia [32,33,34], Iran [23,24], Czechia [35], Indonesia [36], China [37,38,39,40], Malaysia [41,42], and South Korea [43,44], again supporting its applicability globally. This may have been the first study however where the CRBS has been administered in a context without available CR, which did render it infeasible to fully assess construct and criterion validity.

### 4.1. Implications

The CRBS-A is available in anonymous patient-report format online to use open access, with ethics approval: https://globalcardiacrehab.com/Patients-CRBS, accessed on 8 March 2023. Besides the Arabic version, three other languages are also available: English, Portuguese, and Simplified Chinese. After completing the survey, respondents are presented with a list of their top barriers along with the validated and useful suggestions on how to mitigate them, with links to resources in their own language. Respondents can save or print their top barriers or discuss them with healthcare providers. Future research is needed to test if mitigation strategies have an impact on CR utilization using a randomized, controlled, and prospective study design.

As evidenced by the top barriers, structural changes are needed to augment CR utilization; mitigating many of the barriers is beyond the control of patients, such that it is incumbent upon the professional community to mitigate them. Chiefly again, there is a dire need for more CR programs [25], and reimbursement of CR care [45] so that patients can actually be informed about and referred to available programs. For example, satellite CR programs are being initiated in Qatar [46]. Acute cardiac care physicians would then need to be educated about the benefits of CR and the importance of referral [47,48], with the setup of automatic inpatient referral [49,50] so patients do not perceive that their providers consider CR unnecessary [51]. Finally, the 3rd and 4th top barriers were distance and relatedly challenged to in-person sessions, which can be mitigated by hybrid session delivery within CR programs [52]. 

### 4.2. Limitations

While experts from across the EMR gave input, data were only collected from patients in two countries; thus, results may not be generalizable to all Arab-speaking settings. Second, there were multiple comparisons, which can lead to inflated error; the item-level CRBS analysis for criterion validity was solely exploratory and as such, caution is warranted in interpreting the results regarding association to CR enrolment. Third, physical activity history was not assessed using a validated scale. Fourth, causal conclusions cannot be drawn. Finally, future research is recommended to again assess these psychometric properties, and also test others such as test–retest reliability. 

## 5. Conclusions

The CRBS is the only available tool designed to identify cardiac patients’ multi-level barriers to CR enrolment and participation. The Arabic version of this instrument—CRBS-A—was developed and validated in this study. While more research is needed, overall results confirmed good psychometric properties, which supports its administration. Moreover, for the first time, this study has proffered strategies for mitigating them, which providers and patients alike established as helpful. In conjunction with needed health system changes, it is hoped that this freely-available online resource will support mitigation of CR barriers at the patient-level, so ultimately more patients in Arab-speaking countries and beyond reap the life-saving benefits of CR.

## Figures and Tables

**Figure 1 healthcare-11-01196-f001:**
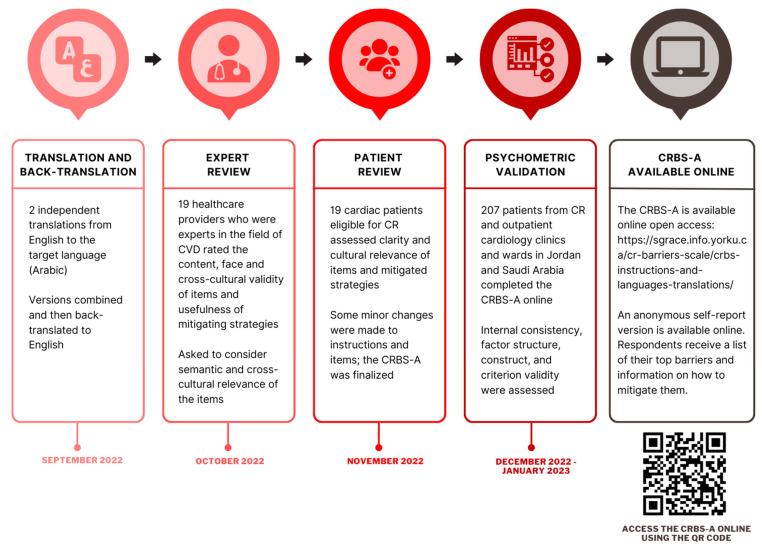
Process of Translating, Adapting and Psychometrically Validating the CRBS-A. Legend: CRBS-A, Cardiac Rehabilitation Barriers Scale-Arabic.

**Table 1 healthcare-11-01196-t001:** Characteristics of validation study participants, and by CRBS-A score (N = 207).

Sociodemographic Characteristics	n (%)/ Mean ± SD	Median CRBS-A Score (Interquartile Range)	*p* ^a^
Age	58.2 ± 13.9	-	0.62
Less than 65 years old	123 (59.4)	2.8 (2.3–3.5)	
65 years old or older	66 (31.9)	2.9 (2.6–3.3)	
Sex			0.27
Male	128 (61.8)	2.9 (2.5–3.4)	
Female	75 (36.2)	2.8 (2.4–3.3)	
Other/prefer not to answer	2 (1.0)		
Work status			0.33
Retired	93 (44.9)	3.0 (2.6–3.4)	
Full or part-time	59 (28.5)	2.7 (2.4–3.5)	
Unemployed	27 (13.0)	2.9 (2.3–3.5)	
Disability/sick leave/modified duties	17 (8.2)	2.7 (2.0–3.1)	
Looking to get a paid job	8 (3.9)	2.7 (2.6–3.3)	
Social support (/5) ^b^	3.3 ± 1.1	-	0.26
Definitely	32 (15.5)	3.1 (2.6–3.5)	
Most of the time	46 (22.2)	3.0 (2.5–3.5)	
Sometimes	78 (37.7)	2.8 (2.4–3.3)	
Rarely	34 (16.4)	2.9 (2.6–3.3)	
Never	13 (6.3)	2.5 (2.1–3.0)	
Years of formal education	13.2 ± 6.2	-	0.66
Less than 8 years	18 (8.7)	2.8 (2.4–3.6)	
8 or more years	152 (73.4)	2.7 (2.4–3.3)	
Worrying about having enough money to pay for health care			0.20
I often worry	65 (31.4)	2.8 (2.5–3.3)	
I worry sometimes	105 (50.7)	3.0 (2.6–3.5)	
I never worry	35 (16.9)	2.7 (2.3–2.9)	
Physical activity prior to heart problem			0.58
Yes	40 (19.3)	2.7 (2.4–3.6)	
No	159 (76.8)	2.9 (2.5–3.4)	
Referral to CR			0.04
Yes	43 (20.8)	2.6 (2.3–3.3)	
No/I do not know	158 (76.3)	3.1 (2.5–3.4)	

CR: cardiac rehabilitation; CBBS-A: Cardiac Rehab Barriers Scale Arabic; SD: standard deviation. a. Spearman’s correlation, Wilcoxon or Kruskal–Wallis test, as appropriate. b. responses were also scored on a Likert scale from 1 “never” to 5 “definitely” so a mean score could be computed.—not applicable.

**Table 2 healthcare-11-01196-t002:** Exploratory factor analysis of the CRBS-A and reliability of factors, N = 207.

Item	Factor 1: Time Conflicts/Lack of Perceived Need/Excuses	Factor 2: Prefer to Self-Manage	Factor 3: Logistical Barriers	Factor 4: Health System Issues/Comorbidities
11. …of time constraints (e.g., too busy, inconvenient class time)	0.799			
7. …I already exercise at home, or in my community	0.768			
12. …of work responsibilities	0.741			
10. …travel (e.g., holidays, business, cottage)	0.688			
6. …I do not need cardiac rehab (e.g., feel well, heart problem treated, not serious)	0.676			
8. …severe weather	0.674			
9. …I find exercise tiring or painful	0.480			
4. … family responsibilities (e.g., caregiving)	0.434			
18. … I can manage my heart problem on my own		0.840		
17. … many people with heart problems do not go, and they are fine		0.824		
19. … I think I was referred, but the rehab program did not contact me		0.786		
21. …I prefer to take care of my health alone, not in a group		0.604		
15. …I am too old		0.572		
3. … difficulties in accessing sessions that require attendance in person (for example, lack of car or suitable transportation)			0.818	
2. … costs (e.g., program participation costs, transportation and parking costs, qualification requirements such as shoes, exercise, equipment/educational materials, equipment costs)			0.725	
1. … distance (e.g., there is not a program in the same area, too far for travel)			0.661	
5. …I did not know about cardiac rehab (e.g., doctor did not tell me about it)				−0.711
16. …my doctor did not feel it was necessary				−0.597
14. …other health problems prevent me from going.				0.581
20. …it took too long to get referred and into the program				0.576
13. …I do not have the energy				0.413
Variance explained	23.3%	17.8%	12.7%	8.9%
Eigenvalues	7.50	2.41	2.03	1.25
Reliability	0.88	0.84	0.78	0.50

**Table 3 healthcare-11-01196-t003:** CRBS-A mean item, factor, and total scores.

CRBS-A Item	Total (N = 207)	N/A (%)
1. …distance (e.g., there is not a program in the same area, too far for travel)	3.4 ± 1.5	21 (10.1)
2. …costs (e.g., program participation costs, transportation and parking costs, qualification requirements such as shoes, exercise, equipment/educational materials, equipment costs)	3.2 ± 1.5	19 (9.2)
3. …difficulties in accessing sessions that require attendance in person (for example, lack of car or suitable transportation)	3.4 ± 1.5	20 (9.7)
4. …family responsibilities (e.g., caregiving)	3.1 ± 1.5	20 (9.7)
5. …I did not know about cardiac rehab (e.g., doctor did not tell me about it)	4.1 ± 1.3	7 (3.4)
6. …I do not need cardiac rehab (e.g., feel well, heart problem treated, not serious)	2.6 ± 1.3	14 (6.8)
7. …I already exercise at home, or in my community	2.6 ± 1.4	17 (8.2)
8. …severe weather	2.8 ± 1.4	16 (7.7)
9. …I find exercise tiring or painful	2.9 ± 1.4	13 (6.3)
10. …travel (e.g., holidays, business, cottage)	2.3 ± 1.5	17 (8.2)
11. …of time constraints (e.g., too busy, inconvenient class time)	2.8 ± 1.4	20 (9.7)
12. …of work responsibilities	2.5 ± 1.6	31 (15.0)
13. …I do not have the energy	2.9 ± 1.4	15 (7.2)
14. …other health problems prevent me from going.	2.2 ± 1.4	33 (15.9)
15. …I am too old	2.7 ± 1.5	21 (10.1)
16. …my doctor did not feel it was necessary	3.6 ± 1.5	10 (4.8)
17. …many people with heart problems do not go, and they are fine	2.6 ± 1.4	17 (8.2)
18. …I can manage my heart problem on my own	2.8 ± 1.4	11 (5.3)
19. …I think I was referred, but the rehab program did not contact me	2.5 ± 1.4	17 (8.2)
20. …it took too long to get referred and into the program	2.5 ± 1.6	28 (13.5)
21. …I prefer to take care of my health alone, not in a group	2.8 ± 1.4	28 (13.5)
Factor 1: time conflicts/lack of perceived need/excuses	2.7 ± 1.0	-
Factor 2: prefer to self-manage	2.7 ± 1.0	-
Factor 3: logistical barriers	3.3 ± 1.3	-
Factor 4: health system issues/comorbidities	3.1 ± 0.8	-
Total CRBS-A	2.9 ± 0.8	-

N/A, not applicable; CRBS-A, CR Barriers Scale-Arabic, CR, cardiac rehabilitation. Note: CRBS-A scores range from 1–5, with higher scores indicating greater barriers. Mean ± standard deviation shown.

## Data Availability

Data available on request due to ethical restrictions.

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
