# Peer review of "Translation, Cross-Cultural Adaptation and Psychometric Validation of the Arabic Version of the Cardiac Rehabilitation Barriers Scale (CRBS-A) with Strategies to Mitigate Barriers"

_healthcare, 2023, doi:10.3390/healthcare11081196_

Round 1
Reviewer 1 Report
Aljehani et al. provided an extensive Arabic traduction of the Cardiac Rehabilitation Barriers Scale (CRBS-A) through an in-depth validation method. The authors should be congratulated for their efforts aiming to extend the accessibility of cardiac rehabilitation across the world. As Antoine de Saint-Exupéry stated: “Language is a source of misunderstanding” Little Prince, Ed. Gallimard 1945. Improvements in the expression of one of the most used tools in the management of cardiovascular disease is certainly a big step in helping the patients and reducing the costs related to miscommunication (re-hospitalization, extensive caregivers consumption…). The article does not need English revisions and is well-written. I have only two minor comments:
- Do the authors mean “Arabic” as a unique language? As European citizen, it is often confusing to understand the difference between Arabic (intended as “scholar” Arabic, as far as I can express it like this), and dialectal/regional Arabic. Is Arabic language from Saudi Arabia different from those spoke in Syria, Iran, or whatever?
- As extension to this work, can the authors comment on the fact that despite an accurate and extensive traduction of such tool, there is no instrument as performant as a person-tailored management by a “real-person” caregiver? A lot of misunderstanding could be avoided if human relation became again central in our current medicine’s practices. It’s honorable to provide such helpful solution, language-based and by the way closer to the patient, but it sounds palliative into a worldwide sick medical system who lacks people, doctors, nurses, social assistant and thereby…who is losing his humanity.
Reviewer 2 Report
I had the opportunity to review this interesting article. The study has provided the Arabic version of CRBS, which has practical implications for future research. Also, the study is well-designed and has a sound methodology. I have a few minor comments, which are pointed out below.
a. Page 1, line 40; MENA and EMR share many countries between them, and they don’t refer to two completely different regions.
b. Page 1, line 41, cardiac rehabilitation is tertiary prevention, not secondary.
c. Page 5, line 191, Please mention the hypotheses you aimed to evaluate in order to assess the scale’s validity.
d. Before using the parametric tests, did you use any normality tests to ensure your variables were distributed normally?
e. How did you evaluate the history of physical activity? Physical activity should be evaluated using standard questionnaires.
f. Page 13, line 17; I think you meant internal consistency rather than internal reliability.
g. I would suggest adding more comparisons between the Arabic and other versions of CRBS to the discussion section.
Reviewer 3 Report
First of all, I want to note that it has been a pleasure review your manuscript. I think this is an interesting work on the translation and cross-culturally adapt the CRBS into Arabic and to solicit patients' feedback on their usefulness.
After reading in depth the manuscript, I would like to make some comments:
- A sentence about the CRBS scale should be added in the introductory part of the introduction to make it clear to the reader what it is about, even if it is explained in more detail later in the materials.
- Line 84 there is too big a gap in the line. Please correct it.
- Lines 114-116. The authors have written: “……community [18]. Using Qualtrics (https://www.qualtrics.com/), they were asked to rate the content validity of items (to enable computation of the content validity index (CVI) the items (I-CVI) and scale (S-CVI) [19] and usefulness of mitigating strategies”.
Why did the authors choose Qualtrics? Are there any studies that have used it before?
- Figure 1 should go in section 2.1 where it is quoted and not as it is placed in section 2.2.
- Line 124.Could the authors please specify a little more about how purposively-sampled was done?
- Line 130-131. The authors commented: …”The target sample size was based on the International Society for Pharmacoeconomics and Outcomes Research (ISPOR) guidelines, which suggest a minimum of 5-8 respondents”.
What was the exact number in this article?
-An introductory paragraph in the results section would be appreciated before starting with the three sub-sections.
- In table 1 in the social support section of the first column what does (/5) mean?
